# Melanoma Cells Inhibit iNKT Cell Functions via PGE2 and IDO1

**DOI:** 10.3390/cancers15133498

**Published:** 2023-07-05

**Authors:** Enza Torre, Giulia Pinton, Grazia Lombardi, Silvia Fallarini

**Affiliations:** Department of Pharmaceutical Sciences, University of Piemonte Orientale, 28100 Novara, Italy; enza.torre@uniupo.it (E.T.);

**Keywords:** iNKT cells, COX2, IDO, immunosuppression

## Abstract

**Simple Summary:**

The unique properties of invariant natural killer T (iNKT) cells make them an attractive candidate for cancer-adoptive immunotherapy. However, despite their potential, clinical studies have not consistently shown successful outcomes. This lack of efficacy is likely attributed to the immunosuppressive nature of the tumor microenvironment. In this study, we investigated the role of melanoma cell lines in suppressing iNKT cell functions, even in the presence of their specific antigen. Additionally, we aimed to identify the key factors responsible for this immunosuppressive effect. Understanding the primary contributors to the failure of iNKT cell-based therapy is crucial for developing new treatment strategies.

**Abstract:**

Invariant natural killer T (iNKT) cells are a distinct group of immune cells known for their immunoregulatory and cytotoxic activities, which are crucial in immune surveillance against tumors. They have been extensively investigated as a potential target for adoptive cell immunotherapy. Despite the initial promise of iNKT cell-based immunotherapy as a treatment for melanoma patients, its effective utilization has unfortunately yielded inconsistent outcomes. The primary cause of this failure is the immunosuppressive tumor microenvironment (TME). In this study, we specifically directed our attention towards melanoma cells, as their roles within the TME remain partially understood and require further elucidation. Methods: We conducted co-culture experiments involving melanoma cell lines and iNKT cells. Results: We demonstrated that melanoma cell lines had a significant impact on the proliferation and functions of iNKT cells. Our findings revealed that co-culture with melanoma cell lines led to a significant impairment in the expression of the NKG2D receptor and cytolytic granules in iNKT cells. Moreover, we observed a strong impairment of their cytotoxic capability induced by the presence of melanoma cells. Furthermore, through the use of selective inhibitors targeting IDO1 and COX-2, we successfully demonstrated that the melanoma cell line’s ability to impair iNKT cell activation and functions was attributed to the up-regulation of IDO1 expression and PGE2 production.

## 1. Introduction

Invariant natural killer T (iNKT) cells are a distinct subset of T cells that exhibit a high degree of evolutionary conservation. They are characterized by the simultaneous expression of natural killer (NK)-lineage receptors and an invariant T-cell receptor (TCR). The TCR typically consists of a Vα24-Jα18 chain paired with a Vβ11 β-chain in humans [1]. Unlike conventional T cells, iNKT cells possess the ability to recognize foreign or self-lipid antigens when presented in the context of the non-polymorphic major histocompatibility complex (MHC) class I-like molecule CD1d [2,3,4].

iNKT cells possess distinctive phenotypic and functional characteristics that position them as regulators and/or effectors in various immune responses. Once activated, iNKT cells have the ability to recruit, activate, and modulate other immune cells through the release of chemokines or cytokines, or by activating receptors. Their cytokine secretion enables them to regulate the suppressive activity of neutrophils, activate macrophages, influence the polarization of macrophages toward M1 or M2 phenotypes, and transactivate natural killer (NK) cells. By expressing CD40, they can stimulate the activation of dendritic cells (DCs) and B cells, as well as enhance the cross-presentation activity of DCs to T cells. Furthermore, iNKT cells can induce cytotoxicity in target cells through two main mechanisms: the death receptor pathway (involving Fas/FasL and DR5/TRAIL), and the release of cytotoxic granules (such as granzyme, perforin, and granulysin) [5,6].

In the context of tumor immunosurveillance, iNKT cells play a crucial role. Smyth et al. reported that mice deficient in iNKT cells showed increased susceptibility to methylcholanthrene-induced sarcomas [7].

The anti-tumoral effect of iNKT cells against tumors relies on both direct and indirect mechanisms. Indirectly, iNKT cells produce Th1 cytokines, including IFN-γ, which promote the proliferation and activation of NK cells, as well as enhance the cytotoxicity of antigen-specific CD8^+^ T cells. Additionally, iNKT cells have the ability to counteract the immunosuppressive activities of pro-tumorigenic myeloid populations such as tumor-associated macrophages (TAMs) and myeloid-derived suppressor cells (MDSC). Directly, iNKT cells can recognize and exhibit cytotoxicity against tumor cells through two alternative pathways. Firstly, iNKT cells can recognize cancer cells that express CD1d molecules via the interaction between the iNKT cell T-cell receptor (TCR) and CD1d. Secondly, iNKT cells can recognize CD1d-negative cancer cells using NK-like mechanisms of target cell recognition, such as NKG2D and TNF receptor ligands [1,7]. The pivotal role of iNKT cells in antitumor immunity makes them an intriguing target for cancer immunotherapy. Numerous studies have reported alterations in iNKT cell numbers and functionality in cancer patients, including reductions in iNKT cell frequency and a shift towards a Th2 phenotype [5,8]. These impairments have been described to correlate with a worse prognosis.

Melanoma represents the most lethal type of skin cancer originating from the malignant transformation of melanocytes. For many years, there were limited therapeutic options available for melanoma patients. However, advancements in our understanding of cancer biology have led to the development of new therapeutic strategies targeting mutated driver genes (e.g., BRAF) and immune checkpoints. The use of monoclonal antibodies against CTLA-4, PD-1, and PD-L1 in this context has improved patient prognosis, although the response rate remains low and the emergence of therapy resistance is a significant concern [9]. Consequently, there is an urgent need to explore new therapeutic approaches to broaden treatment options for melanoma patients.

Focusing on the potential of iNKT cells to exert antitumor activities in melanoma patients, several immunotherapeutic approaches have been proposed. In animal models, interventions such as the administration of α-galactosyl ceramide (α-GC) or α-GC-loaded dendritic cells (DCs), as well as the adoptive transfer of activated iNKT cells, have resulted in an increased frequency of iNKT cells and robust cytokine responses. These findings suggest the restoration of iNKT cell antitumor function [10,11]. Clinical trials conducted to evaluate the efficacy of iNKT cell-based immunotherapies in melanoma patients have yielded inconsistent results, partially attributed to the development of tumor immune escape mechanisms. Tumor cells are known to release various cytokines (such as TGF-β, IL-10), mediators (including kynurenine, PGE2, etc.), and to recruit immunosuppressive cells (such as TAMs and MDSCs) that contribute to immune evasion [12]. The effective manipulation of iNKT cells in antitumor therapeutics heavily relies on a comprehensive understanding of the signals that negatively regulate iNKT cells in cancer and the signals capable of unleashing their potential antitumor activity [1]. Currently, limited data are available regarding the negative regulation imposed by melanoma cells on iNKT cells. This understanding is pivotal in identifying subsequent immuno-therapeutic approaches capable of unleashing the antitumor activity of iNKT cells against melanoma cells.

The findings presented in this study demonstrate that melanoma cell lines directly regulate the activity of iNKT cells, resulting in a reduction in both the regulatory and effector functions of iNKT cells. This regulation is characterized by a shift in the produced cytokines towards the Th2 type and a decrease in the expression of receptors and cytotoxic machinery necessary for eliminating tumor cells.

## 2. Materials and Methods

### 2.1. Monoclonal Antibodies and Flow Cytometric Analysis

The monoclonal antibodies (mAb) used in this study were anti-Vα24Vβ11 (6B11, 10 μL for flow cytometry tests), anti-perforin (dG9), anti-granzyme (CB6), anti-FasL (MFL3), anti-CD107a (1D4B), anti-CD8 (SK1), anti-CD4 (GK1.5), anti-CD3 (17A2), and anti-NKG2d (1D11), which were purchased from Biolegend, Dan Diego, USA (5 μL for flow cytometry tests). For cytofluorimetric analysis, cells were stained with the appropriate mAb, washed, and stained with the appropriate isotype-specific mAb. For intracellular staining, cells were fixed (4% paraformaldehyde), permeabilized (0.1% saponin) (Sigma-Aldrich, Milan, Italy), and stained with the appropriate mAb. All samples were analyzed using a FACSVantage-SE^®^ flow cytometer (BD Biosciences, Milan, Italy). To compare the surface density and the intracellular content of cytotoxic granules among iNKT cells cultured in different conditions, we express them as median fluorescent intensity (MFI). Data analysis was conducted using FACSDiva software (BD Biosciences).

### 2.2. Cell Culture

The human A375 and WM 266-4 melanoma cell lines were maintained in DMEM, supplemented with 10% fetal bovine serum (FBS), 2 mM L-glutamine, 100 U/mL penicillin, and 100 mg/mL streptomycin (GE Healthcare, Milan, Italy).

Before co-culture experiments, melanoma cell line proliferation was inhibited by mitomycin treatment. A375 and WM 266-4 cell lines were treated with 0.5 mg/mL of the antimitotic mitomycin C (Sigma-Aldrich) for 1 h and 30 min, respectively, washed with PBS, detached with trypsin/EDTA, counted and plated at defined density.

The human CD1d-transfected THP-1 cell line, kindly provided by Prof. Gennaro De Libero (Department of Research, Experimental Immunology, University Hospital Basel, Basel, CH), was regularly cultured in RPMI-1640 complete growth medium, containing 10% heat-inactivated FBS, 100 μg/mL kanamycin, 1 mM sodium pyruvate, 2 mM L-glutamime, 1% MEM amino acid solution, and 0.01 mM β-mercaptoethanol (GE Healthcare).

Peripheral blood mononuclear cells (PBMC) were obtained from healthy volunteers after informed consent and separated by Ficoll density gradient centrifugation (Histopaque 1077, Sigma-Aldrich). Human iNKT cells were expanded from resting PBMC by α-GC treatment [13]. Briefly, resting PBMC (5 × 10^6^ cells/mL) were treated with 100 ng/mL of α-GC (KRN7000; kindly provided by Prof. Luigi Panza, “Dipartimento di Scienze del Farmaco”, University of “Piemonte Orientale”, Novara, Italy) for 12 days. Subsequently 20 U/mL of human recombinant IL-2 (rhIL-2) (Peprotech, New York, NY, USA) was added after 48 h, followed by 40 U/mL of IL-2 every 2 days until the end of cell proliferation. At day 12, human iNKT cells were positively selected using anti-human Vα24Vβ11 conjugated microbeads (Miltenyi Biotec, Milan, Italy). iNKT cells were routinely purified to >97% using this method. Isolated iNKT cells were cultured in supplemented RPMI-1640 with 100 U/mL of IL-2 before use.

All cells were maintained at 37 °C in a humidified atmosphere of 95% air and 5% CO_2_ until use.

### 2.3. Proliferation of iNKT Cells

To evaluate the effect of the melanoma cell line on iNKT cell proliferation, coculture experiments were performed. PBMC (1 × 10^6^ cells/mL per ml) were labeled with 0.25 μM carboxyfluorescein succinimidyl ester (CFSE) (Life Technologies, Monza, Italy) in serum-free PBS for 30 min at 37 °C. FBS was added to stop the reaction and the cells were washed several times with RPMI-1640. CFSE-labeled PBMCs (2 × 10^6^ CFSE-labeled PBMC) were stimulated with 100 ng/mL of α-GC and co-cultured with A375 or WM 266-4 cells at different melanoma/PBMC ratios (1:20 and 1:40) for 12 days. In total, 20 U/mL of rhIL-2 was added after 48 h, followed by 40 U/mL of rhIL-2 every two days until the end of cell proliferation. The controls were α-GC-treated PBMC. On day 12, PBMC were harvested, stained with anti-Vα24Vβ11, anti-CD8, and anti-CD4 antibodies, and analyzed using FACS.

In some experiments addressing the role of IDO1 or COX-2 in melanoma cell immunomodulation, increasing concentrations (10–100 μM) of the IDO1 inhibitor methyl-tryptophan (1MT) or 10 μM of the COX-2 inhibitor (rofecoxib) (Sigma-Aldrich, Milan, Italy) were added at the onset of co-cultures. Transwell experiments were performed by plating melanoma cells in the upper chamber and α-GC-treated PBMC in the lower chamber of a 0.4 μm transwell plate.

The effect of L-kynurenine on iNKT cell proliferation was also investigated. CFSE-labeled PBMC were stimulated with 100 ng/mL α-GC and rhIL-2 in the presence or absence of increasing concentrations (1–100 μM) of L-kynurenine (Sigma-Aldrich). On day 12, PBMC was harvested, stained with anti-Vα24Vβ11, and analyzed using FACS.

### 2.4. iNKT Cell/Melanoma Cell Co-Culture

Purified iNKT cells were cultured in RPMI medium containing rhIL-2 (100 U/mL), either in the absence or presence of mitomycin C-treated melanoma cell lines at different melanoma/iNKT cell ratios (1:2 and 1:10) for four days. When required, 100 μM of the IDO1 inhibitor 1MT or 10 μM of COX-2-inhibitor rofecoxib were added at the onset of the co-culture.

### 2.5. Degranulation Assay

iNKT cells were stimulated as described in the “*iNKT cell/melanoma cell co-culture*” section. Harvested iNKT cells were incubated with α-GC unloaded/loaded THP1 CD1d at 1:1 E:T ratio for 1 h at 37 °C in the presence of anti-CD107a. After 1 h Monensin (GolgiStop reagent) (BD Biosciences, Milan Italy) was added, and cells were incubated for 3 h at 37 °C. Cells were then harvested washed and analyzed using FACS.

The effect of l-kynurenine on iNKT cell degranulation was investigated. iNKT cells were cultured in RPMI medium and rhIL-2 100 U/mL in the presence/absence of increasing concentrations (1–100 μM) of L-kynurenine. On day 4, iNKT cell degranulation was performed, as previously reported.

### 2.6. FACS Analysis of iNKT Cell Cytotoxic Mediators

iNKT cells stimulated as described in the “*iNKT cell/melanoma cell co-culture”* section were harvested, washed, and labeled with anti-Vα24Vβ11, anti-perforin, anti-granzyme B, anti-FasL and anti NKG2d mAbs and analyzed by FACS (BD Biosciences). Non-specific background fluorescence was evaluated with the appropriate isotype-matched control mAb. Perforin, graznyme B, FasL, α24Vβ11-TCR, and NKG2D levels were expressed as MFI by using FACSDiva software (BD Biosciences).

### 2.7. Cytokine Secretion Assays

iNKT cells were co-cultured as described in the “*iNKT cell/melanoma cell co-culture”* section. After 48 h of co-culture, cell media were collected and stored at −20 °C until the day of analysis. To assess the ability of conditioned iNKT cells to respond to α-GC stimulation, co-cultured iNKT cells were harvested and stimulated with α-GC loaded THP1 CD1d. After 48 h the culture media were collected and stored at −20 °C until the analysis. The levels of IFN-γ and IL-4 released in the cell culture media were assessed using enzyme-linked immunosorbent assay (ELISA) kits (BioLegend, San Diego, CA, USA) according to the manufacturer’s instructions. The concentrations of IFN-γ and IL-4 in the samples were determined by extrapolation from the specific reference standard curves.

### 2.8. Cytotoxicity Assays

Cytotoxicity assays were performed using Calcein-AM (CAM) (Life Technologies). A total of 1 μM of CAM was added to target cells, and the cells were incubated at 37 °C for 15 min. After washing, labeled target cells were seeded in a 96-well plate at a density of 5 × 10^4^ cells in 50 μ µL per well. iNKT cells, stimulated as described in the “*iNKT cell/melanoma cell co-culture*” section, were harvested, diluted and 50 μL was added per well to reach the desired E:T ratio. Each plate included only the target cells as controls for spontaneous cell death measurements. The plate was incubated at 37 °C in a humidified atmosphere with 5% CO_2_ for 5 or 24 h. After incubation, the cells in each well were harvested, washed, and labeled with propidium iodide (PI), and cytotoxicity was measured using FACS. Live target cells were identified as CAM^high^/PI^−^ populations, whereas dead target cells were CAM^low^/PI^+^ and the effector iNKT cells were CAM^−^ (at least 10-fold less fluorescent than killed target cells). After gating the target cells, cytotoxicity was calculated as the % increase in the CAM^low^/PI^+^ population relative to the target cells alone (cytotoxicity, % = [CAM^low^/PI^+^ in experimental wells—CAM^low^/PI^+^ in control wells] /CAM^high^/PI^−^ in control wells × 100). The mean cytotoxicity % SEM for each condition was calculated from three replicate experimental wells.

### 2.9. Reverse Transcriptase-Polymerase Chain Reaction (RT-PCR)

RNA extraction and RT-PCR analysis were performed as previously described [14]. Briefly, A375 and WM 266-4 were co-cultured with iNKT cells at different E:T ratios, in the absence/presence of transwell, for 4 days, harvested, and centrifuged at 1000× *g* for 5 min at 4 °C. Total RNA was isolated using the GenElute™ mammalian total RNA miniprep kit (Sigma-Aldrich) and reverse-transcribed using the ThermoScript™ RT–PCR kit (Life Technologies) according to the manufacturer’s instructions. For amplification, 3 μL of cDNA was added to GoTaq FlexiDNA Polymerase in 25 μL reaction buffer, containing 0.5 mM of forward and reverse primers (Appendix A). RT-PCR amplicons were resolved in a 1% agarose gel by electrophoresis, and signals were quantified with densitometric analysis software (NIH Image 1.32; National Institutes of Health, Bethesda, MD, USA). Data were expressed as the ratio of the signals obtained for each gene in one sample divided by that obtained for the reference gene (human GAPDH) in the same sample.

### 2.10. Determination of IDO Enzymatic Activity

The enzymatic activity of IDO was evaluated by measuring the levels of L-kynurenine in co-culture media, as previously described [15]. The co-cultures were executed as reported above using a complete growth medium, with a 100 μM (f.c.) L-tryptophan (Sigma-Aldrich). Amounts of L-kynurenine in cell media were quantified on the basis of a calibration curve. The detection limit of this method was 1 μM.

### 2.11. PGE_2_ Measurement

A375 and WM 266-4 cell lines were untreated/treated with 10 μM rofecoxib. PGE_2_ levels in culture supernatants were determined using a commercially available enzyme-linked immunosorbent assay kit (R&D Systems, Minneapolis, MN, USA) according to the manufacturer’s recommendations.

### 2.12. Statistical Analysis

The results are expressed as mean ± SEM of at least four experiments. Statistical significance was evaluated using a one-way ANOVA followed by Student’s *t*-test for paired populations using GraphPad Prism 9 (GraphPad Software, Inc., San Diego, CA, USA). Differences were considered statistically significant at *p* < 0.05. Data were fitted as sigmoidal concentration–response curves and analyzed with a four-parameter logistic equation using GraphPad Prism 9 software (GraphPad Software, Inc., San Diego, CA, USA).

## 3. Results

### 3.1. Melanoma Cell Lines Inhibit α-Galcer Induced iNKT Cell Proliferation

Co-cultures of iNKT cells and A375 or WM 266-4 melanoma cell lines were established in order to investigate the potential effects of melanoma cells on iNKT cells. Since iNKT cells are present at low frequencies in the peripheral blood of healthy donors (with a mean value of 0.2% in PBMC from eight donors), the iNKT cell population was expanded in culture from PBMC for 12 days in the presence or not of A375 and WM 266-4 melanoma cell lines at 1:20 and 1:40 effector (E): target (T) ratios in the presence of α-galactosylceramide (αGC). Consistent with previously published findings [16], our study demonstrated that 12 days of α-galactosylceramide (αGC) stimulation resulted in a selective expansion of the initial iNKT cell population. The percentage of Vα24Jα18-Vβ11+ cells increased from 0.2 ± 0.1% in PBMC to 16.4 ± 0.5% in αGC-stimulated PBMC. As shown in Figure 1A, co-culturing iNKT cells with both melanoma cell lines significantly inhibited iNKT cell proliferation at all E:T ratios tested (*p* ≤ 0.05). Notably, the A375 cell line exhibited a higher inhibitory effect, resulting in a 90% reduction in iNKT cell proliferation in αGC-treated PBMC at a 1:20 E:T ratio (Figure 1B).

To assess whether the presence of tumor cells influenced the proliferation of specific subsets (CD8^+^, CD4^+^, DP, and DN) of iNKT cells, we collected iNKT cells after 12 days of expansion, both in the presence and absence of melanoma cell lines. The collected iNKT cells were labeled with specific antibodies and analyzed using flow cytometry (FACS) to evaluate their phenotype. The phenotypic analysis, as shown in Figure 1C, demonstrated that the presence of tumor cell lines led to a significant increase in the CD8^+^ subtype (*p* ≤ 0.05) and a slight reduction in the percentage of the CD4^+^ subtype. However, the percentages of DN and DP subsets remained unchanged (Figure 1C). To investigate whether the inhibition of iNKT cell expansion was mediated by soluble factors released by melanoma cells, we conducted experiments using a transwell system. As depicted in Figure 1D, melanoma cells exhibited a significant reduction in iNKT cell expansion (*p* ≤ 0.05), albeit to a lesser extent compared to the reduction observed without the transwell (Figure 1D). This suggests that both cell-cell contact and soluble factors play a role in the inhibitory effect of melanoma cells on iNKT cell expansion. These results suggest that the reduction in iNKT cell proliferation is likely attributable to soluble factors released by melanoma cell lines. To test this hypothesis, we cultured iNKT cells with media obtained from melanoma cell lines and conditioned media from αGC-treated PBMC/melanoma cell co-cultures. As depicted in Figure 1E, the media obtained from αGC-treated PBMC/melanoma cell co-cultures resulted in a reduction in iNKT cell proliferation.

### 3.2. Melanoma Cell Lines Affect iNKT Cell Effector Functions

To investigate whether melanoma cells impact the antitumor effector functions of iNKT cells, we conducted co-culture experiments with iNKT cells co-cultured for 4 days with A375 or WM 266-4 melanoma cell lines at specific E:T ratios. The expression of CD107a, a degranulation marker associated with both cytokine and cytotoxic granule release, was then evaluated. We utilized αGC-unloaded/loaded THP-1 CD1d cells as triggering cells in this analysis. As depicted in Figure 2A, iNKT cells exhibited a partial response when stimulated with unloaded THP-1 CD1d cells, and a higher degranulation response when stimulated with αGC-loaded THP-1 CD1d cells (Figure 2B). However, when co-cultured with melanoma cells at a 1:2 ratio, iNKT cells demonstrated a diminished ability to be triggered by both αGC-unloaded (Figure 2A) and αGC-loaded THP-1 CD1d cells (Figure 2B). Notably, when co-cultured at a 1:10 melanoma/iNKT cell ratio, the function of iNKT cells was preserved.

To determine whether the reduced degranulation of iNKT cells affected cytokine production, we stimulated iNKT cells with αGC-unloaded/loaded THP-1 CD1d cells and analyzed the supernatant for secreted IFN-γ and IL-4 levels after 4 days of culture, in the presence/absence of melanoma cells. As depicted in Figure 2C, iNKT cells produced detectable levels of IFN-γ and IL-4 in the presence of THP-1 unloaded cells. Furthermore, the levels of IFN-γ and IL-4 were increased when iNKT cells were stimulated with αGC-loaded THP-1 CD1d cells (Figure 2D). Upon co-culture with melanoma cells, iNKT cells stimulated with THP-1 unloaded cells exhibited defective IFN-γ production at a 1:2 melanoma/iNKT cell ratio (Figure 2C). However, the effect on IL-4 production differed depending on the co-culture with A375 or WM 266-4 cells. Specifically, the presence of WM 266-4 cells resulted in an increase in IL-4 production, while the presence of A375 cells had no significant effect. When iNKT cells were co-cultured with melanoma cell lines and stimulated with αGC-loaded THP-1 CD1d cells, the production of IFN-γ was significantly (*p* ≤ 0.05) reduced at both E:T ratios tested (Figure 2D). Similarly, the effect on IL-4 production varied between the co-cultures with A375 or WM 266-4 cells, with WM 266-4 cells inducing an increase in IL-4.

We further analyzed whether the cytotoxic activity of iNKT cells was impaired by co-culture with melanoma cells. As iNKT cells can kill tumor cells via NKG2D-dependent or TCR-dependent mechanisms, we investigated whether melanoma cells could affect both of these pathways. We performed cytotoxicity experiments using αGC-loaded THP1 cells to assess TCR-dependent iNKT cell cytotoxicity and unloaded THP1 cells to assess TCR-independent iNKT cell cytotoxicity. As shown in Figure 3A,B, iNKT cells exhibited cytotoxic activity against both unloaded and αGC-loaded THP-1 CD1d cells, although with different efficiencies. However, their cytotoxic activity against both unloaded and αGC-loaded THP-1 CD1d cells was significantly reduced when co-cultured with melanoma cells (Figure 3A,B). The strongest effect was observed in iNKT cells co-cultured with A375 cells.

Next, we assessed whether the functional defects observed in iNKT cells co-cultured with melanoma cells were reflected in the altered expression of certain iNKT cell surface receptors (such as TCR, NKG2D, and FasL) and molecules involved in iNKT cell cytotoxic activity (such as granzymes and perforins). After 4 days of co-culture with melanoma cells, iNKT cells were collected and characterized using immunofluorescence and FACS analyses to examine the expression of cytotoxic receptors (FasL and NKG2D) as well as the presence of major components of cytolytic granules (granzymes and perforins). The presence of melanoma cells significantly affected the expression of NKG2D, granzyme, and perforin, albeit with varying degrees of efficacy. However, analysis of FasL expression indicated that while the surface density remained unchanged, the percentage of FasL-positive cells increased upon co-culture with melanoma cells. Furthermore, the expression of TCR was not affected by melanoma cells (Figure 3C,D). Taken together, these results demonstrate that melanoma cells exert an inhibitory effect on both the regulatory and cytotoxic effector functions of iNKT cells.

### 3.3. IDO1 and COX-2 Are Involved in the Inhibition of iNKT Cell Proliferation

As shown in Figure 1, the inhibitory effects of melanoma cells on iNKT cell proliferation were mediated by soluble factors released by the melanoma cells. Previous studies [12,13] have demonstrated that IDO1 and COX-2 can impair the proliferation and function of immune cells, such as NK cells and T cells. Therefore, we investigated the expression of IDO1 and COX-2 in our experimental model.

Neither the WM 266-4 nor the A375 cell lines constitutively expressed IDO1 mRNA. However, the expression of IDO1 was induced by IFN-γ in both cell lines (Figure 4A). Interestingly, as shown in Figure 4B, the melanoma cell lines expressed IDO1 mRNA even when co-cultured with iNKT cells at both E:T ratios tested. To determine whether the released IFN-γ by activated iNKT cells played a role in the induction of IDO1 expression in melanoma cells, we co-cultured iNKT cells with A375 or WM 266-4 cells in the presence or absence of an anti-IFN-γ blocking antibody. The addition of the anti-IFN-γ antibody to melanoma/iNKT cell co-cultures completely abolished the expression of IDO1 in both melanoma cell lines, as shown in Figure 4C.

In contrast to IDO1, which showed inducible expression, both melanoma cell lines exhibited high basal levels of COX-2 expression (Figure 4D).

As depicted in Figure 5, the expression of COX-2 in melanoma cell lines correlated with the release of PGE2 in the culture medium. Both WM 266-4 and A375 cells produced PGE2, with the highest production observed in WM 266-4 cells (5505 ± 504 pg/mL). Treatment with rofecoxib, a COX-2 inhibitor, significantly and strongly reduced the production of PGE2 in the culture media by approximately 98%. This suggests that the observed high levels of PGE2 are predominantly produced through the enzymatic activity of COX-2 in melanoma cells.

### 3.4. L-Kynurenine Produced by IDO1 Enzymatic Activity Inhibits iNKT Cell Proliferation

We tested whether IDO1 expressed by melanoma cells co-cultured with iNKT cells was enzymatically active. To determine its activation, we measured the amount of L-kynurenine in co-culture supernatants. As shown in Figure 5, supernatants obtained from melanoma cells did not contain L-kynurenine. However, it was detected in the supernatants from both cell lines co-cultured with αGC-treated PBMC (Figure 6A). The greatest increase in L-kynurenine concentrations was observed in the A375/αGC-treated PBMC co-culture. To assess whether the inhibitory effect of iNKT cell proliferation was due to L-kynurenine produced via IDO1, we analyzed its direct effect on iNKT cell proliferation. As reported in Figure 6B, the addition of L-kynurenine to the culture medium reduced iNKT cell proliferation in a concentration-dependent manner. Notably, inhibition of iNKT cell proliferation was observed at L-kynurenine concentrations comparable to those measured in the co-culture systems.

### 3.5. Released L-Kynurenine and COX-2 Activity Are Responsible for Melanoma Mediated Inhibion fo iNKT Cell Proliferation

We further investigated whether IDO1 and COX-2 activity is responsible for the inhibition of iNKT cell proliferation mediated by melanoma cells. αGC-treated PBMC were co-cultured with melanoma cell lines in the presence or absence of increasing concentrations of IDO1 and COX-2 inhibitors (1MT and rofecoxib, respectively). iNKT cell proliferation and L-kynurenine levels in the co-culture medium were evaluated. The treatment with 1MT partially restored iNKT cell proliferation in a concentration-dependent manner (Figure 7A,B), and, as expected, the concomitant reduction in L-kynurenine levels (Appendix A) confirmed that IDO1 was involved in the melanoma-mediated reduction in iNKT cell proliferation through the production of L-kynurenine. Furthermore, treatment with a specific COX-2 inhibitor prevented the reduction in the proliferation of both cell lines in a concentration-dependent manner.

### 3.6. IDO1 and COX-2 Activities Inhibits iNKT Cell Activity

Consistently with the role of IDO1/L-kynurenine and COX-2, we tested their impact on iNKT cell effector functions, including cytokine production and cytotoxic activity. iNKT cells were cultured alone or co-cultured with WM 266-4 or A375 cell lines in the presence or absence of 1MT and rofecoxib. After 4 days of co-culture, supernatants were collected and tested for levels of IFN-γ and IL-4. Additionally, iNKT cells were harvested and assessed for degranulation and cytotoxicity properties. iNKT cells were cultured for 4 days in the presence or absence of increasing concentrations (1–100 μM) of L-kynurenine, and degranulation was evaluated. As shown in Appendix A, treatment with L-kynurenine affected iNKT cell degranulation in a concentration-dependent manner. Analysis of IFN-γ and IL-4 levels in the culture media of iNKT cells co-cultured for 4 days with melanoma cells in the presence of inhibitors revealed that 1MT and rofecoxib partially restored the release of both cytokines (Figure 8A,B and Appendix A).

Moreover, we demonstrated that the addition of 1MT or rofecoxib to melanoma/iNKT cell co-cultures partially restored both NKG2D-mediated (Figure 9A) and TCR-mediated iNKT cell cytotoxicity (Figure 8B). The restoration of iNKT cell cytotoxicity was more pronounced in the presence of rofecoxib (Figure 8B), indicating that COX-2 activity had a stronger inhibitory effect on the cytotoxic function of iNKT cells. Furthermore, as expected, the greatest increase in iNKT cell cytotoxicity in the presence of rofecoxib was observed in the iNKT cell co-culture with WM 266-4 cells, which express a higher basal level of COX-2. The observed effect on iNKT cell cytotoxicity positively correlated with the partial prevention of melanoma-mediated downregulation of NKG2D, perforin, and granzyme by 1MT and rofecoxib treatment (Figure 9C). These functional data strongly suggest that IDO1 and COX-2 play important roles in the immunosuppressive activity of melanoma cells against iNKT cell activity.

## 4. Discussion

We reported that the inhibitory effect exerted by melanoma cells was mainly due to COX-2 activity and IDO1 expression, resulting in the release of IFN-γ from activated iNKT cells.

iNKT cells play a critical role in tumor immunosurveillance and contribute to antitumor immune responses. In patients with hematologic and solid tumors, reduced frequency and function of intratumoral or circulating iNKT cells have been observed [17,18]. Furthermore, a decrease in iNKT cell frequency correlates with poor overall survival in tumor patients, whereas increased numbers of iNKT cells have been associated with a better prognosis [19]. The causes of iNKT cell impairment are not fully understood, but understanding them is crucial for effectively utilizing iNKT cells in cell-based immunotherapy.

It is well known that various components of the tumor microenvironment can affect the number and function of iNKT cells. It has been established that: (i) circulating myeloid-derived DCs from cancer patients reduce activation and Th1 cytokine production in an IL-10 and TGF-β-dependent manner in iNKT cells [20]; (ii) TAMs reduce both iNKT cell proliferation and cytokine production in a PD-1-dependent manner [21]; (iii) Tregs reduce immunosurveillance by decreasing iNKT cell number and cytotoxic activity [22]; and (iv) MDSCs suppress iNKT antitumor activity [23]. In contrast, the direct contribution of tumor cells to the suppression of iNKT cell number and activity is less well defined. Limited data are available regarding the effects of lung cancer cell lines on the cytokine profiles of CD8^+^ NKT cells [24]. To address this knowledge gap, in this study we characterized the impact of melanoma cell lines on iNKT cells.

A common characteristic observed in cancer patients, regardless of tumor type, is a deficiency in the number of iNKT cells and their proliferative response upon TCR engagement [5], which is associated with a poor prognosis [25]. This reduction could be attributed to increased iNKT cell death, recruitment of iNKT cells to the tumor site, or impaired iNKT cell proliferation. Within the tumor microenvironment, tumor cells can negatively impact the activity, proliferation, and viability of T and NK cells through various mechanisms such as nutrient depletion, pH reduction, hypoxia, metabolites, cytokines (e.g., lactic acid, kynurenines, IL-10, TGF-β, PGE2, and TNF-α), and modulation of surface molecule expression (e.g., CTLA-4, PD-L1, and reduced expression of costimulatory molecules) [12,26].

We provide the first evidence that melanoma cells interfere with iNKT cell proliferation and effector functions, leading to a diminished capability of iNKT cells to directly kill tumor cells and regulate antitumor immune responses. Through co-culture and transwell experiments, we demonstrated that both the immune-regulatory and cytotoxic activities of iNKT cells against tumors are affected by melanoma cells. Our findings suggest that both direct interactions and soluble factors released by melanoma cells play a role in this impairment. Indeed, we observed a reduction in iNKT cell proliferation even when they were cultured in a conditioned medium obtained from melanoma cells.

In our experimental model, we observed a decrease in IFN-γ levels and an increase in IL-4 production. These findings strongly suggest that the reduced production of IFN-γ by iNKT cells observed in cancer patients [27] is likely to be partially attributable to the direct effect of tumor cells. The decrease in IFN-γ, a key cytokine produced by iNKT cells, has significant implications for their role in orchestrating antitumor immune responses. IFN-γ secretion by iNKT cells plays a crucial role in activating and recruiting both innate and adaptive immune cells. It can transactivate NK cells, enhancing NK-mediated tumor cell cytolysis. Additionally, in vivo, iNKT cells can promote DC maturation, leading to increased expression of MHC II and co-stimulatory molecules, as well as potent IL-12 production, which further supports IFN-γ secretion by iNKT cells. Both IL-12 and IFN-γ are important drivers for the priming and activation of tumor antigen-specific CD8^+^ T cells [9]. Our data demonstrate a shift in the cytokine balance towards a Th2 phenotype in iNKT cells co-cultured with WM 266-4 melanoma cells. This suggests that melanoma cells not only diminish the antitumor immune responses of iNKT cells but also promote immune tolerance. It is well-established that IL-4 is a tumor-promoting molecule, as many tumor cells express high levels of the IL-4 receptor. Elevated levels of IL-4 protect tumor cells from apoptosis and contribute to the establishment and maintenance of Th2-polarized immune responses. This, in turn, reduces the cytotoxic activity of CD8^+^ T cells against tumor cells and indirectly impairs antitumor immunity in both tumor-bearing mice and cancer patients. Furthermore, some studies indicate that cancer cells have the ability to stimulate Th2-type cytokines, including IL-4, to evade the immune system. Additionally, when tested in a DC/iNKT cell model, certain cancer cell extracts have been shown to induce an increase in IL-4 levels [28,29,30].

As previously described, tumor cells have a direct impact on the cytotoxicity of antigen-specific CD8^+^ T cells and NK cells. In this study, we have demonstrated the ability of melanoma cells to diminish iNKT cell cytotoxicity by down-regulating the expression of perforin, granzyme, and NKG2D, while increasing the percentage of iNKT cells expressing FasL. These findings indicate that melanoma cells can evade iNKT cell cytotoxicity not only by down-regulating the surface expression of CD1d but also through alternative pathways [9].

In our experimental model, we observed that the inhibitory effect on iNKT cells occurred even in the absence of direct contact between iNKT cells and tumor cells. This effect was most pronounced after co-culturing iNKT cells with tumor cells, suggesting that it could be attributed to the release of soluble factors by melanoma cells and other factors triggered by the interaction between iNKT cells and tumor cells.

In the tumor microenvironment (TME), various immune cells, including NK cells, T cells, and NKT cells, produce IFN-γ, which plays a crucial role in immune activation. IFN-γ induces NK cell activation and recruits innate and adaptive immune cells to the tumor site, promoting their activation. However, extensive research has shown that IFN-γ can also contribute to tumor initiation and facilitate changes in the phenotype of tumor cells, promoting their growth in immunocompetent hosts. IFN-γ can induce the expression of more than 200 genes, including those involved in immune evasion by tumor cells, such as PD-L1, PD-L2, CTLA-4, CIITA, non-classical MHC class Ib antigens, IDO1, and CXCL12 [30].

IDO1, in particular, plays a crucial role in suppressing antitumor immune responses by producing immunosuppressive molecules. IFN-γ promotes the expression of IDO1 in primary and metastatic tumor cells, intratumoral endothelial cells, immune cells in the peritumoral stroma, and tumor-draining lymph nodes [31]. IDO1 activity leads to the degradation of tryptophan and the production of kynurenines, which have several immunosuppressive effects. These include inhibiting mTOR1 and activating GCN2, resulting in the inhibition of T-effector cell function and apoptosis, suppressing NK cell proliferation and function, promoting the differentiation and activation of regulatory T (Treg) cells, tolerogenic dendritic cells (DCs), and myeloid-derived suppressor cells (MDSCs), as well as contributing to tumor neovascularization [32].

In our co-culture model of tumor cells with iNKT cells, we observed IFN-γ-dependent induction of IDO1 expression in both tumor cell lines, albeit at different levels. The kynurenines produced as a result of IDO1 activity directly inhibit iNKT cell function. However, we demonstrated that the inhibitory effects on iNKT cells can be partially restored in the presence of the IDO1 inhibitor 1MT [33]. This clearly demonstrates that IDO1 expression and activity in tumor cells contribute to the impairment of iNKT cell functions induced by the tumor. Additionally, our findings indicate the involvement of another key player in mediating these effects. In recent years, there has been a growing understanding of the relationship between chronic inflammation and cancer development, with a particular focus on the role of COX-2. COX-2 expression has been found to be higher in various premalignant and malignant epithelial cell lesions, indicating its involvement in local immune suppression [34]. Numerous in vitro and in vivo studies have demonstrated that tumor-derived PGE2, produced by COX-2, plays a significant role in promoting angiogenesis [35] and suppressing antitumor immune responses.

PGE2 has been shown to inhibit macrophage differentiation into the M1 phenotype, which is associated with antitumor activity, and impairs the ability of T and NK cells to effectively kill tumor cells [36]. Additionally, PGE2 can induce the polarization of Th and macrophage cells toward pro-tumorigenic phenotypes, specifically Th2 and M2 phenotypes, respectively [37,38,39]. These phenotypic shifts contribute to an immunosuppressive microenvironment that favors tumor growth and progression.

The findings from these studies emphasize the importance of COX-2 and its downstream product PGE2 in shaping the tumor microenvironment and suppressing antitumor immunity. Targeting COX-2 and its associated pathways may hold promise as a therapeutic strategy to counteract the immunosuppressive effects of chronic inflammation in cancer.

Our findings align with previous studies demonstrating that COX-2 plays a similar inhibitory role in iNKT cells, compromising their regulatory and cytotoxic antitumor functions. Similar to NK cells, the activity of both IDO1 and COX-2 leads to a decrease in the expression of the activating receptor NKG2D on the surface of iNKT cells, as well as a reduction in the levels of intracellular granzyme and perforin, which are essential for recognizing and eliminating tumor cells. The involvement of PGE2 in inducing iNKT cell anergy has been previously documented in murine inflammatory models. In these models, PGE2 delivered by intestine-derived exosomes during inflammatory conditions or produced by Kupffer cells during viral hepatitis infection has been shown to exert inhibitory effects on liver-resident iNKT cells stimulated with α-GalCer. These effects include decreased proliferative responses, cytokine release, and expression of activation markers [40,41]. RT-PCR analysis has revealed an upregulation of both EP2 and EP4 receptors in α-GalCer-stimulated iNKT cells, and treatment with specific EP2 and EP4 antagonists has been shown to reduce PGE2-induced anergy, indicating the involvement of both receptors in mediating the observed inhibitory effects. Interestingly, the EP4 antagonist appears to have a more pronounced effect [42,43]. Taken together, these findings strongly suggest that PGE2 plays a critical role in inducing anergy in iNKT cells, impairing their functional capabilities. This further underscores the significance of the IDO1 and COX-2 pathways, including PGE2 production, in modulating the immune response mediated by iNKT cells in the context of cancer.

Further in vivo studies in the murine melanoma cancer model will be useful to better elucidate iNKT anergy-inducing mechanisms in order to boost anti-tumor immunity and provide protection from tumor progression.

## 5. Conclusions

In conclusion, our findings shed light on additional factors that contribute to the immunosuppression of iNKT cells. Furthermore, they propose potential therapeutic approaches by targeting IDO1 and PGE2 pathways. Inhibitors of IDO1 and antagonists of PGE2 could serve as promising adjuvants in combination with immunotherapeutic strategies, aiming to fully restore the antitumor immune functions of iNKT cells. These novel therapeutic strategies have the potential to enhance the efficacy of immunotherapy and bolster the antitumor immune responses mediated by iNKT cells.

## Figures and Tables

**Figure 1 cancers-15-03498-f001:**
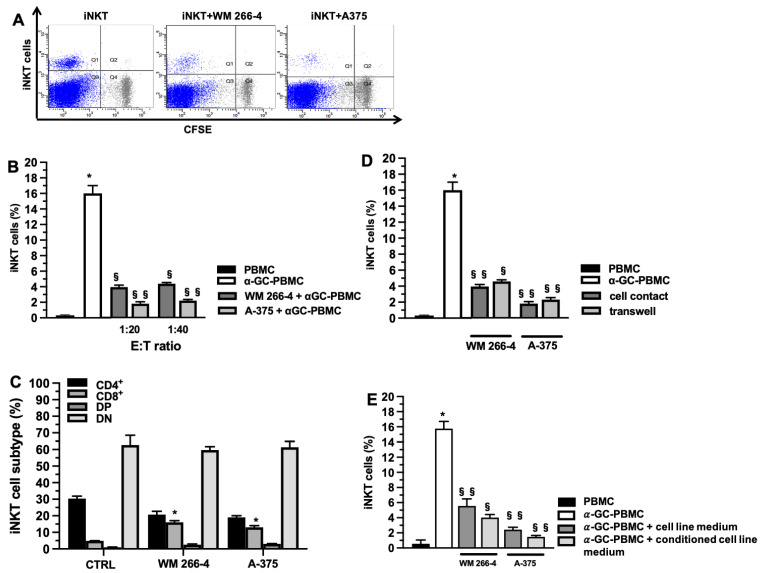
Melanoma cell lines impair iNKT cell proliferation. (**A**) Representative dot plots of αGC-mediated iNKT cell proliferation (12 days) measured as CFSE fluorescent intensity reduction due to cell division in the absence (left panel), or presence of WM 266-4 or A375 melanoma cell lines (middle and right panel respectively); (**B**) percentages of αGC-mediated iNKT cell proliferation (6 days) in absence/presence of melanoma cell lines at different E:T ratio; (**C**) percentages of proliferated iNKT cell subsets (CD8^+^, CD4^+^, DP, and DN) in absence/presence of melanoma cell lines; (**D**) percentages of αGC-mediated iNKT cell proliferation (12 days) in cell contact or transwell system in absence/presence of melanoma cell lines; (**E**) percentages of αGC-mediated iNKT cell proliferation (12 days) in presence of melanoma cell medium or conditioned melanoma cell medium (medium obtained by αGC-treated PBMC/melanoma cell co-cultures). Results are represented as mean ± SEM of at least 6 experiments conducted with PBMC isolated from different healthy volunteers. * *p* ≤ 0.05 vs. αGC-unstimulated PBMC; § *p* ≤ 0.05, §§ *p* ≤ 0.01 vs. αGC-stimulated PBMC in absences of melanoma cell lines.

**Figure 2 cancers-15-03498-f002:**
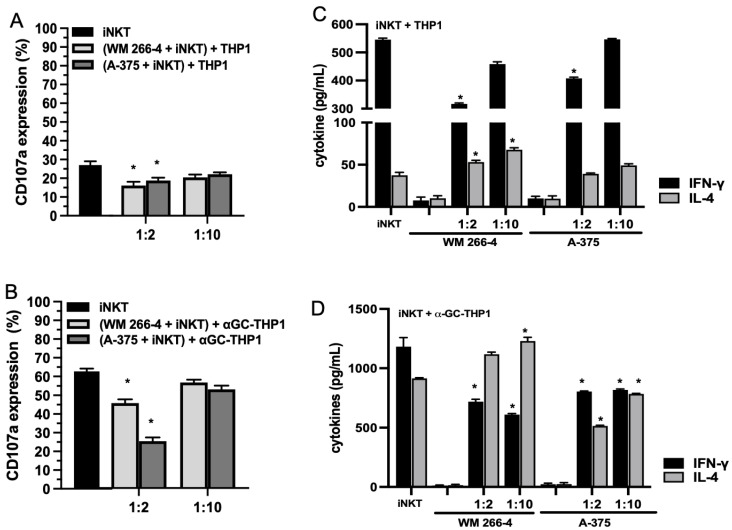
Degranulation and cytokine production are reduced in iNKT cells co-cultured with melanoma cell lines. (**A**,**B**) iNKT cells cultured 4 days in the absence/presence of melanoma cell lines (different E:T ratio) were analyzed for CD107a expression in redirected degranulation assay against THP-1 unloaded/loaded with αGC; (**C**,**D**) IFN-γ and IL-4 levels were measured by ELISA on surnatants from iNKT cells cultured 4 days in absence/presence of melanoma cell lines (different E:T ratio) and stimulated with THP-1 unloaded/loaded with αGC for 48 h. Results are represented as mean ± SEM of at least 6 experiments conducted with iNKT cells isolated from different healthy volunteers. * *p* ≤ 0.05 vs. iNKT cultured in absences of melanoma cell lines.

**Figure 3 cancers-15-03498-f003:**
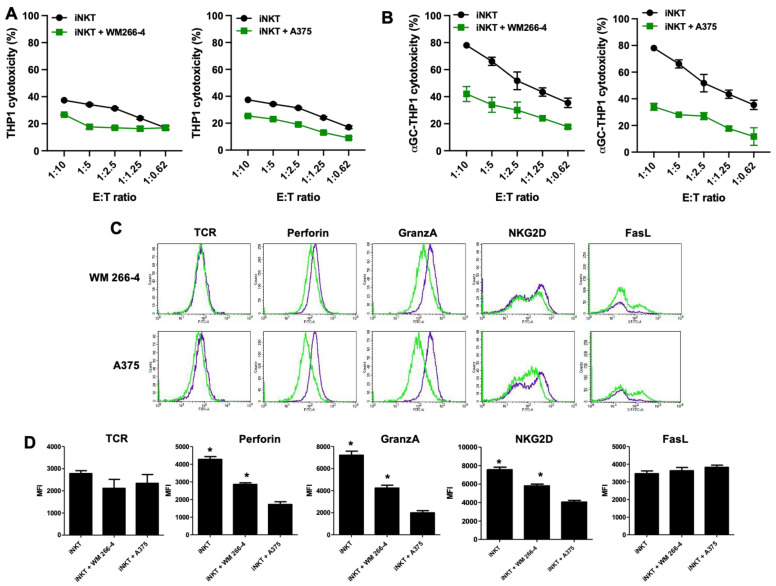
Cytolytic functions are impaired in iNKT cells co-cultured with melanoma cell lines. (**A**,**B**) iNKT cells cultured 4 days in the absence/presence of melanoma cell lines were analyzed for cytolytic activity in redirected killing assay against THP-1 unloaded/loaded with αGC; (**C**) the expression of receptors/molecules involved in cytolytic activity on iNKT cells cultured 4 days in the absence (purple lines) or presence (green lines) of melanoma cell lines were analyzed by flow cytometry. A representative experiment out of 6 conducted is shown; (**D**) expression level of receptors/molecules involved in cytolytic activity on iNKT cells cultured 6 days in the absence or presence of melanoma cell lines. Results are represented as mean ± SEM of at least 6 experiments conducted with iNKT cells isolated from different healthy volunteers. * *p* ≤ 0.05 vs. iNKT cultured in absences of melanoma cell lines.

**Figure 4 cancers-15-03498-f004:**
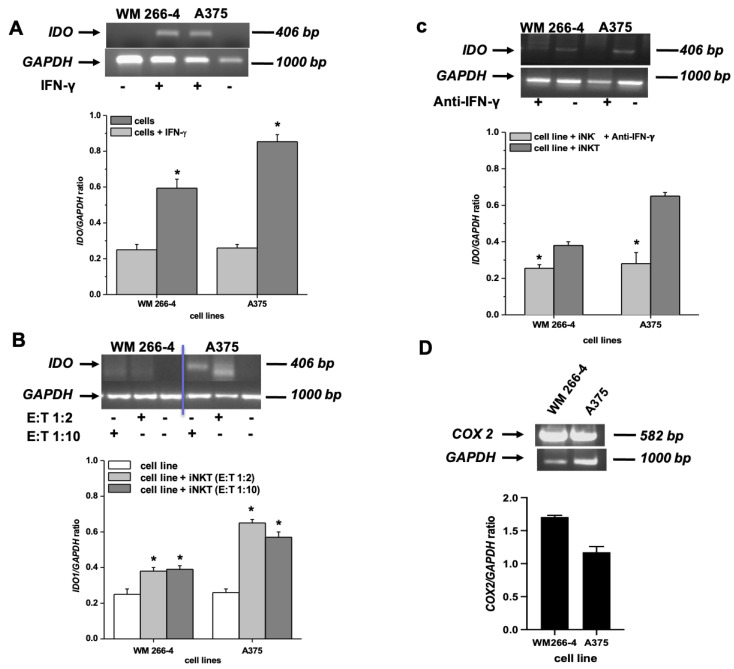
Expression of IDO and COX-2 mRNA by melanoma cell lines. (**A**–**C)** IDO mRNA expression was assessed by RT-PCR on melanoma cell lines: treated with IFN-γ, co-cultured for 96 h with iNKT cells at different E:T ratios in the presence/absence of anti-IFN-γ blocking antibody; (**D**) COX-2 expression was assessed by RT-PCR on melanoma cell lines. * *p* ≤ 0.05 vs. iNKT cultured in absences of melanoma cell lines. Original western blots presented in File S1.

**Figure 5 cancers-15-03498-f005:**
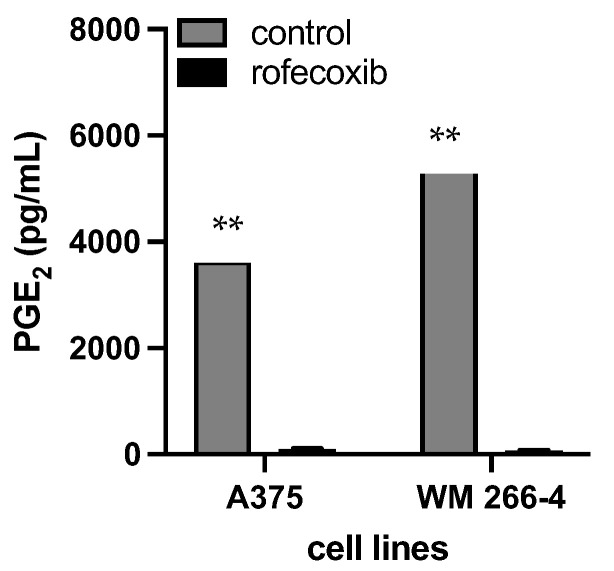
Level of released PGE_2_ in the presence/absence of COX-2 inhibitor. Media form A375 and WM 266-4 untreated/treated with 10 μM of rofecoxib 24 h were harvested and PGE2 levels evaluated by ELISA assay. ** *p* ≤ 0.01 vs. melanoma cell cultures in the absence of rofecoxib.

**Figure 6 cancers-15-03498-f006:**
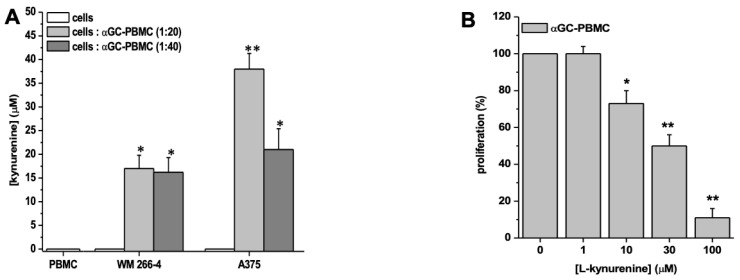
Kynurenine production by melanoma cell lines and iNKT proliferation in the presence of kynurenine. (**A**) Kynurenine concentration in surnatants from αGC-stimulated PBMC cultured in the absence/presence of melanoma cell lines at different E:T ratios was analyzed by HPLC; (**B**) Results are represented as mean ± SEM of at least 6 experiments conducted with PBMC isolated from different healthy volunteers. * *p* ≤ 0.05, ** *p* ≤ 0.01 vs. αGC-stimulated PBMC cultured in absences of melanoma cell lines.

**Figure 7 cancers-15-03498-f007:**
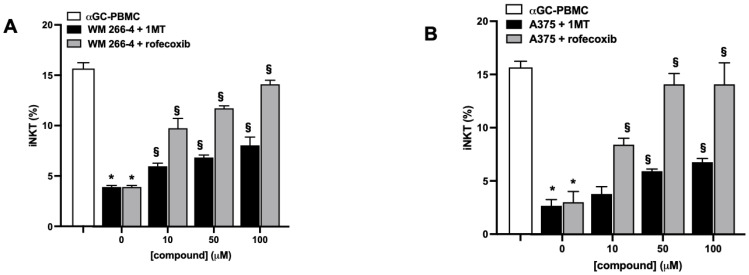
IDO1 or COX-2 activities inhibit iNKT cell proliferative responses to αGC. (**A**,**B**) PBMC stimulated (12 days) with αGC in the presence/absence of melanoma cell lines were treated with increasing concentrations (0–100 μM) of IDO1 or COX-2 inhibitors (1MT and rofecoxib, respectively) and iNKT cell proliferation analyzed by FACS. Results are represented as mean ± SEM of at least 6 experiments conducted with PBMC or iNKT cells isolated from different healthy volunteers. * *p* ≤ 0.05 vs. αGC-stimulated PBMC or iNKT cells cultured in absences of melanoma cell lines; § *p* ≤ 0.05 vs. inhibitors untreated co-cultures.

**Figure 8 cancers-15-03498-f008:**
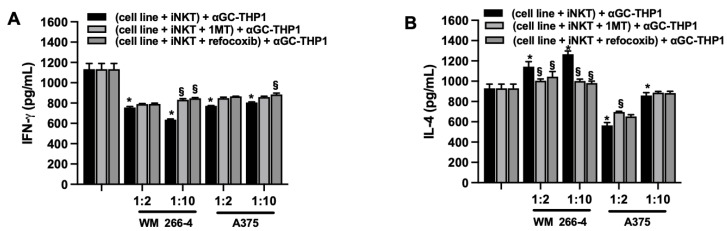
IDO1 or COX-2 inhibit iNKT cell degranulation. (**A**) IFN-γ and (**B**) IL-4 levels were measured by ELISA on surnatants from iNKT cells co-cultured (4 days) without/with melanoma cell lines (different E:T ratio) in the presence of IDO1 or COX-2 inhibitors (1MT and rofecoxib, respectively) and stimulated (48 h) with αGC-loaded THP-1. Results are represented as mean ± SEM of at least 6 experiments conducted with PBMC or iNKT cells isolated from different healthy volunteers. * *p* ≤ 0.05 vs. iNKT cultured in absences of melanoma cell lines. § *p* ≤ 0.05 vs. inhibitors untreated co-cultures.

**Figure 9 cancers-15-03498-f009:**
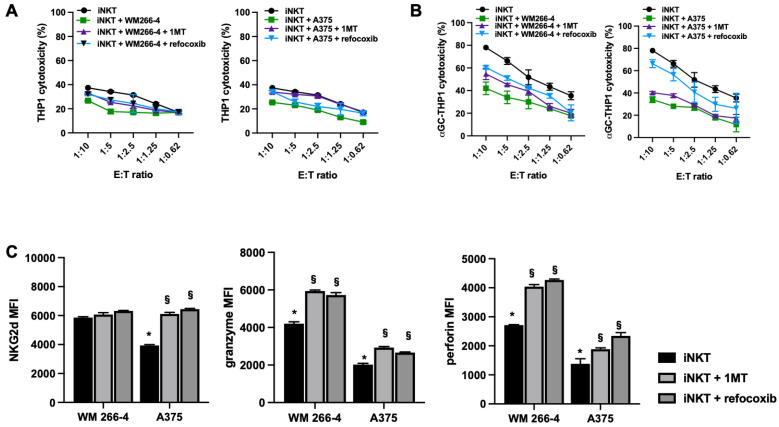
IDO1 or COX-2 impair iNKT cell cytotoxicity and cytolytic mediator expression. (**A**,**B**) iNKT cell co-cultured (4 days) without/with melanoma cell lines in the presence of IDO1 or COX-2 inhibitors (1MT and rofecoxib, respectively) were analyzed for cytolytic activity in redirected killing assay against THP-1 unloaded/loaded with αGC; (**C**) expression of receptor/molecules involved in cytolytic activity on iNKT cells co-cultured (4 days) without/with of melanoma cell lines in presences of IDO1 or COX-2 inhibitors (1MT and rofecoxib, respectively) were analyzed by FACS. * *p* ≤ 0.05 vs. iNKT cultured in absences of melanoma cell lines. § *p* ≤ 0.05 vs. inhibitors untreated co-cultures.

## Data Availability

The data generated in this study are available upon request from the corresponding author.

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
