# Peer review of "Melanoma Cells Inhibit iNKT Cell Functions via PGE2 and IDO1"

_cancers, 2023, doi:10.3390/cancers15133498_

Round 1

Reviewer 1 Report

The authors studied an interesting topic. The results are meaningful and well-documented. Some differences are minimal, even though statistically significant. Thus, the authors should be more drawn while formulating conclusions. Also in the abstract be more careful while underlying their findings, which are based on the cell cultures and not on the native melanoma cells. 

The language is understandable, but I suggest reviewing the manuscript by a native English speaker. 

Author Response

Thus, the authors should be more drawn while formulating conclusions.

            According with the reviewer suggestion we have modified the part of the paper changing the melanoma cells with melanoma cell lines to be less general in our conclusions.

Reviewer 2 Report

In the work by Torre et al, entitled “Melanoma cells inhibit iNKT cell functions via PGE2 and IDO1”. The authors show that melanoma cells affect iNKT cell proliferation and function. They do so via reduction of NKG2D, granzyme, perforines, etc, likely via IDO1 and COX-2 activity. Overall, this is an interesting and mostly well designed paper that aims at understanding how this cell population could be rendered active again to exert its anti-tumor activity.

However, there are several points that need clarification prior to acceptance in Cancers.

1)    Fig. 2A: the difference between 1:2 and 1:10 co-culture ratio is minimal. unclear how it is significant at 1:2.

2)    There is no melanoma only control. Do these cells produce cytokines? And if so which ones and in what amount? This is a co-culture, and even though the rest of the data suggest IFNg and IL4 are predominantly coming from iNKT, a melanoma only control should be added. This could also explain why IL4 is more in co-culture than less?. Also, a cytokine array would be good to pinpoint the overall change in cytokines with and without melanoma cells (including a control of melanoma cells only)

3)    Fig 3D: please label each graph with the name of the marker measured.

4)    Fig. 4C: anti-IFNg in the gel appears to increase IDO1 in both melanoma lines; yet, the graph below does not represent it in any way. And in the results the very opposite is described. Present a blot that accurately represent the quantification, or vice versa. At this point, because of the serious discrepancy between the data and the text, please provide clear proof that anti-IFNg does inhibit IDO expression (multiple original blots with no cutting, etc).

5)    Fig. 4B: IDO1 in WM266-4 not very convincing. Why is the IDO band lower in A375 + E:T 1:2? Finally, this looks like two separate gels. Either repeat on one gel or clearly demarcate with a line, for example, that these are two separate gels.

6)     Fig. 4D: Cox2 levels seem clearly lower in A375, however, later in the paper, the author states “Moreover as expected the major increase in iNKT cell cytotoxicity in presence of rofecoxib was observed for iNKT cell co-culture with A375 cells that express 442 a higher basal level of COX-2.” Based on the data, this is not true.

7)    Fig 6: label graphs better

8)    Fig 7B: in WM266-4, the co-culture induces IL4. Explain

9)    Lines 492-94: “we reported that iNKT cell proliferation was reduced also when iNKT cells proliferation were cultured in the conditioned medium by melanoma cells”. Where are these data?

10) finally, given the good results in vitro obtained with the IDO and COx2 inhibitors, an in vivo experiment with tumor growth measurements, and TME analysis would nicely complete this story.

English is overall good, only minor typos.

Author Response

Reviewer #1

1) Fig. 2A: the difference between … unclear how it is significant at 1:2.

I agree with the revisor that the differences between the 1:2 and 1:10 ratio are minimal but when I apply the student t test the significatively doesn’t come.

2) There is no melanoma only control.

In realty the control sample with the cell line alone were done every time. Since the absorbances read were as the blank we have decided to don’t put they inside the graphs. Since in Reviewer opinion these data lead clearer results, we have now added they in the respective graphs. New Fig. 2C and D

3) Fig 3D: please label each graph with the name of the marker measured.

In the new version of the figure all the label required have been added; see new Fig. 3D

4) Fig. 4C: anti-IFNg in the gel appears to increase IDO1 in both melanoma lines…

I apologize for the mistake in the Fic 4C. During the preparation of the PCR results I have reversed the plus and minus signs. In this version I have correct the mistakes as you can see in the new figure 4C

5) Fig. 4B: this looks like two separate gels. Either repeat on one gel or clearly demarcate with a line, for example, that these are two separate gels…

The samples were run in the same gel but are not in these sequences so I have to cut the bands within the presented results. As the revisor suggests I have now put a line between the two parts.

6) Fig. 4D: Cox2 levels seem clearly lower in A375...

I have corrected the text in the new version, according with the R evisor observation. Pag 13 line 448

7) Fig 6: label graphs better

The graph label has been improved as required. See the new fig 6

8) Fig 7B: in WM266-4, the co-culture induces IL4. Explain

The possible explanation for the IL-4 production by iNKT cells in WM 266-4 co-culture is explained in the discussion section page 15 lines 516-527

9) Lines 492-94: Where are these data?

This result is reported in the Fig 1 and in the results section paragraph 3.1 “Melanoma cell lines inhibit a-galcer induced iNKT cell proliferation” lines 271-280

10) finally, given the good results in vitro in vivo experiment complete this story.

I agree with the revisor that in vivo experiments can offer a more complete view of the interplay between cancer cells and iNKT cells in a real context, and represent the better way to demonstrate the role of the suggested IDO and COX2 pathways in the iNKT cell activity inhibition. Unfortunately, carrying out these experiments is outside of my competence.

Reviewer 3 Report

This is a nice manuscript systematically showing the effect of melanoma cancer cells' immune suppression through inhibition of  invariant natural killer T (iNKT) cells' function in co-culture in vitro models. They further reveals that this inhibition is mediated by COX-2/PGE2 axis and IDO1. overall a clear and nicely written paper.

My only suggestion is that since hypothesis claims COX-2 activation and PGE2 secretion are the main mediators of this effect, It's important to show PGE2 secretion levels in the media by ELISA.

Author Response

It's important to show PGE2 secretion levels in the media by ELISA.

            According to the reviewer suggestion we have performed and ELISA assay to evaluate the PGE2 levels in cell culture media of the used cell lines in presence/absence of the specific inhibitor rofecoxib.